# A cortical disinhibitory circuit for enhancing adult plasticity

Yu Fu[1]*[†], Megumi Kaneko[1][†], Yunshuo Tang[2,3,4], Arturo Alvarez-Buylla[2,3], Michael P Stryker[1]*

[1]Center for Integrative Neuroscience, Department of Physiology, University of California, San Francisco, San Francisco, United States; [2]Department of Neurological Surgery, University of California, San Francisco, San Francisco, United States; [3]The Eli and Edythe Broad Center of Regeneration Medicine, University of California, San Francisco, San Francisco, United States; [4]Medical Scientist Training Program, Biomedical Science Program, University of California, San Francisco, San Francisco, United States

**Abstract** The adult brain continues to learn and can recover from injury, but the elements and operation of the neural circuits responsible for this plasticity are not known. In previous work, we have shown that locomotion dramatically enhances neural activity in the visual cortex (V1) of the mouse (*Niell and Stryker, 2010*), identified the cortical circuit responsible for this enhancement (*Fu et al., 2014*), and shown that locomotion also dramatically enhances adult plasticity (*Kaneko and Stryker, 2014*). The circuit that is responsible for enhancing neural activity in the visual cortex contains both vasoactive intestinal peptide (VIP) and somatostatin (SST) neurons (*Fu et al., 2014*). Here, we ask whether this VIP-SST circuit enhances plasticity directly, independent of locomotion and aerobic activity. Optogenetic activation or genetic blockade of this circuit reveals that it is both necessary and sufficient for rapidly increasing V1 cortical responses following manipulation of visual experience in adult mice. These findings reveal a disinhibitory circuit that regulates adult cortical plasticity.

*For correspondence: yufu@phy. ucsf.edu (YF); stryker@phy.ucsf. edu (MPS)

[†]These authors contributed equally to this work

Competing interests: The authors declare that no competing interests exist.

Cortical plasticity declines with aging, accounting for decreased learning and memory, as well as reduced neural rehabilitation in aging brain (*Singer, 1995*; *Park and Reuter-Lorenz, 2009*). Running or other physical exercise has been suggested to improve many aspects of brain function in aging human beings, including brain plasticity (*Voss et al., 2013*). In aged animals, environmental enrichment has also been shown to improve learning and memory, as well as cortical plasticity; but the underlying circuit mechanisms are unknown (*Vivar et al., 2013*; *Greifzu et al., 2014*).

Our laboratory recently showed that running enhances both visual cortical responses and plasticity in adult mice (*Niell and Stryker, 2010*; *Kaneko and Stryker, 2014*). We also found that running potently activates VIP neurons in mouse primary visual cortex (V1), which in turn inhibit SST inhibitory neurons, thereby disinhibiting the excitatory pyramidal neurons and allowing them to respond more strongly to the visual stimuli for which they are selective (*Fu et al., 2014*). We also showed that activating VIP neurons is both sufficient and necessary for enhancing visual responses during running (*Fu et al., 2014*). Therefore, we set out to investigate whether the VIP-SST disinhibitory circuit, rather than general aerobic exercise, is responsible for enhanced cortical plasticity in adult mice.

To examine the function of VIP neurons in enhancing adult plasticity by running, we silenced their synaptic transmission in binocular zone of mouse V1 by injecting AAV-DIO-TetanusToxinLightChain-T2A-GFP (AAV-DIO-TeTx) into the VIP-Cre mice (*Figure 1A*) (*Xu and Sudhof, 2013*). We then compared the visual responses of stationary (running speed < 2 cm/s) and running (running speed > 5 cm/s)

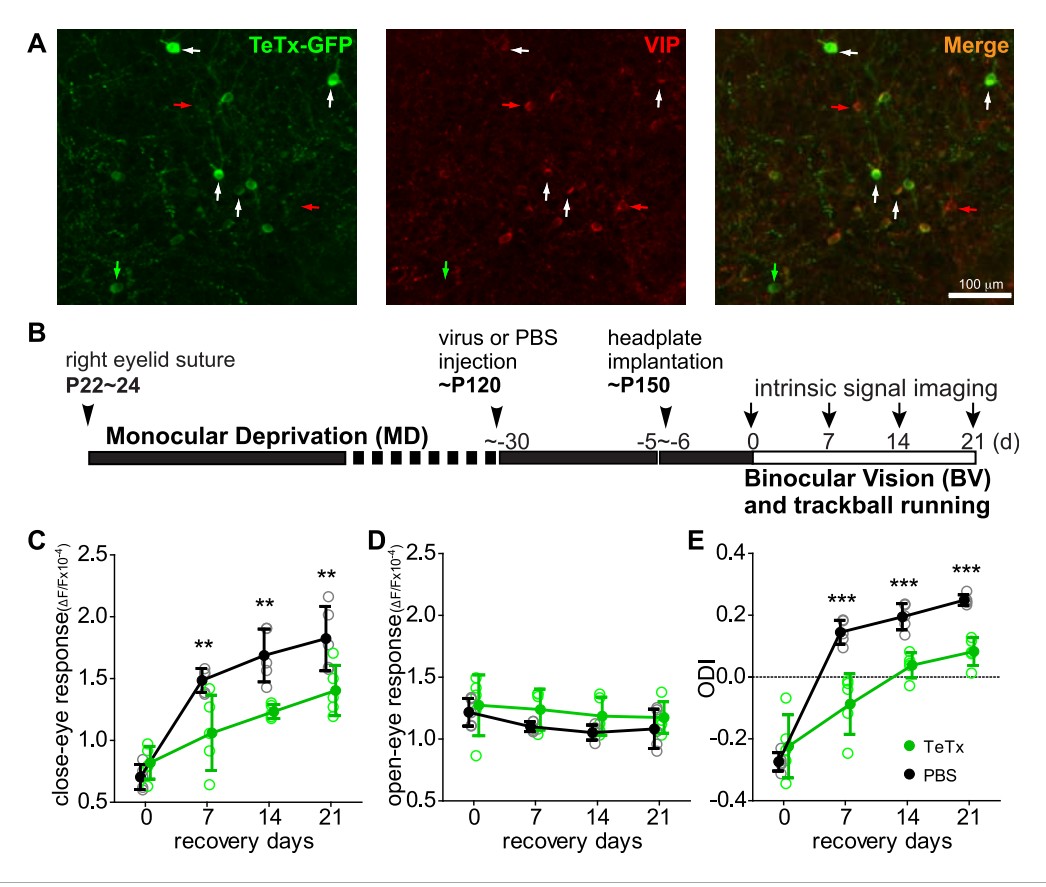

Figure 1. Synaptic transmission from VIP neurons is necessary for the enhancement of recovery of the amblyopic eye by running. (**A**) Representative fluorescent images of binocular V1 area from a VIP-Cre mouse injected with AAV-DIO-TeTx-GFP. Slices were immunostained for GFP to indicate viral infected neurons and VIP for VIP-positive neurons. White arrows indicate the cells positive for both GFP and VIP staining. Red arrows indicate the cells stained positive for VIP only; green arrow indicates a cell positive only for GFP. (**B**) Experimental schedule. (**C** and **D**) Changes in intrinsic signal responses evoked by the visual stimulation through the closed eye (**C**) and the open eye (**D**) in AAV-DIO-TeTx-injected (VIP-TeTx, n = 5) experimental and PBS-injected control mice (PBS, n = 6). (**E**) Ocular dominance index (ODI) computed from responses to contralateral (closed) and ipsilateral (open) eyes shown in (**C** and **D**). ODI represents normalized difference in response magnitude between the two eyes; higher ODI indicates more domination of the contralateral eye. Open circles represent measurements in individual animals, solid circles indicate mean of the open circles. (Data are plotted as mean ± S.D., ***p < 0.001, **p < 0.01 between groups at given time point; Two-way ANOVA followed by multiple comparisons with Bonferroni correction).

The following figure supplements are available for figure 1:

**Figure supplement 1**. Silencing the synaptic transmission of VIP neurons abolished the effect of running on visually responsive neurons.

**Figure supplement 2**. VIP-Cre mice injected with AAV-DIO-TeTx-GFP (TeTx) ran similarly to PBS-injected control mice (PBS).

states, and found no significant difference (increase of peak response at preferred orientations: 4 ± 33%, mean ± SD, p = 0.74) (*Figure 1—figure supplement 1*), indicating that silencing the transmission of VIP neurons by TeTx abolished the effect of running on visually responsive V1 neurons, consistent with our previous report (*Fu et al., 2014*).

We previously showed that running in conjunction with visual stimulation enhanced visual cortical plasticity in a mouse model of human amblyopia (*Kaneko and Stryker, 2014*). To test the requirement

of VIP-cell synaptic transmission in this model, we first sutured closed one eyelid of VIP-Cre mice at ~P24, before the peak of critical period, and then after 4 months injected 1 μl of AAV-DIO-TeTx or PBS (the vehicle solution for virus) into the binocular zone of V1. After an additional month, we re-opened the closed eye and allowed the animals to run on Styrofoam balls floating on air while viewing a visual stimulus 4 hr per day for 21 days, as described previously (*Kaneko and Stryker, 2014*). Cortical responses through two eyes were measured by an experimenter blind to the treatment with intrinsic signal imaging immediately after eye opening and every 7 days during the recovery period (*Figure 1B*). Intrinsic signal imaging provides a reliable and non-invasive measurement of visual cortical responses to the two eyes, permitting long-term repeated measurement of the same animal to assess plasticity (*Kaneko et al., 2008*). Its results have been extensively validated by electrophysiological recordings in previous studies (*Kaneko et al., 2008*; *Kaneko and Stryker, 2014*). As expected, the deprived-eye visual responses recovered well in the control PBS-injected VIP-Cre mice, and the ocular dominance index (ODI) reached a level that is similar to non-deprived animals (*Figure 1C–E*, black traces, compare to *Figure 1* of *Kaneko and Stryker (2014)*). In contrast, V1 responses to the deprived eye in the AAV-DIO-TeTx injected animals recovered only modestly, even after 21 days (PBS: $1.82 \pm 0.26$, TeTx: $1.40 \pm 0.20$, $p = 0.006$), with an ODI significantly smaller than in the control animals (PBS: $0.249 \pm 0.07$, TeTx: $0.083 \pm 0.02$, $p = 0.0004$; *Figure 1C–E*, green traces). Indeed, the poor recovery of AAV-DIO-TeTx injected animals was similar to that of animals that did not run in our previous report (*Kaneko and Stryker, 2014*), despite the fact that they ran as much as PBS-injected animals (% of running time: TeTx, $67.88 \pm 23.6$ vs PBS, $68.14 \pm 17.57$; average running velocity in cm/s: TeTx, $10.78 \pm 3.142$ vs PBS, $9.43 \pm 5.29$; *Figure 1—figure supplement 2*). In these animals, as in all of the TeTx-treated animals described below, we noted no obvious changes in temperament or visual behavior; experimental and control animals appeared indistinguishable. Responses of the open-eye were not significantly changed in either group of animals (*Figure 1D*). Silencing the transmission of VIP neurons thus abolished the effect of running on adult plasticity in this mouse model of amblyopia by reducing the potentiation of deprived-eye responses during recovery.

During the critical period, 3 days of monocular deprivation (MD) is always sufficient to shift the ocular dominance by reducing the deprived-eye response. In adult mice (>3 months old), 7 days of MD is required for a reliable shift of the ocular dominance, and this shift is produced by increasing the open-eye responses (*Sato and Stryker, 2008*). Ocular dominance plasticity in adult mice is thus not only much slower but is also qualitatively different from that during the critical period (*Sato and Stryker, 2008*). To determine whether locomotion would enhance another measure of adult plasticity, we examined changes in visual responses after short-term MD in adult mice that had been reared normally (*Figure 2A*). As expected, 4-day MD did not significantly change the ODI or the response of either eye in adult animals housed in standard conditions (*Figure 2*, 'B6 Home cage' group). However, 4-day MD combined with daily visual stimulation and running led to enhanced plasticity, including both increases in open-eye responses and decreases in deprived-eye responses (ODI before and after MD: $0.25 \pm 0.03$ vs $0.09 \pm 0.08$, $p < 0.001$; closed-eye response before and after MD: $2.37 \pm 0.28$ vs $1.95 \pm 0.34$, $p < 0.01$; open-eye response before and after MD: $1.41 \pm 0.18$ vs $1.69 \pm 0.25$, $p < 0.01$; *Figure 2*, 'B6 VS + run' group). To examine the role of VIP-cell synaptic transmission in the effect of running on this measure of adult plasticity, we injected AAV-DIO-TeTx into the binocular V1 of adult VIP-Cre mice that had been reared normally and measured the changes in visual responses after 4-day MD combined with daily visual stimulation and running (*Figure 2A*). After silencing the synaptic transmission of VIP neurons, running in conjunction with visual stimulation still led to significant depression of the closed-eye response ($2.51 \pm 0.31$ vs $2.23 \pm 0.29$, $p < 0.05$; *Figure 2*, 'VIP-TeTx VS + run' group) but failed to potentiate the open-eye response ($1.51 \pm 0.20$ vs $1.53 \pm 0.18$, $p > 0.05$; *Figure 2C*, 'VIP-TeTx VS + run' group), resulting in a much smaller change of ODI after 4D-MD ($0.25 \pm 0.03$ vs $0.18 \pm 0.05$, $p < 0.05$; *Figure 2D*, 'VIP-TeTx VS + run' group). These results indicate that the VIP neurons are specifically involved in potentiating the adult-form plasticity.

We next investigated whether activating VIP neurons would be sufficient to enhance adult plasticity even without the exercise of daily running. We expressed a variant of channelrhodopsin, ChETA, in VIP neurons of binocular V1 by infecting VIP-Cre mice with AAV-DIO-ChETA-YFP (*Figure 3A*); we had previously shown this treatment to enhance visual responses of excitatory cells in stationary mice upon stimulation with blue light (*Fu et al., 2014*). We imaged baseline visual responses ~2 weeks after viral injection. After implanting the fiber optic cannula in the binocular zone of V1, we sutured one eyelid

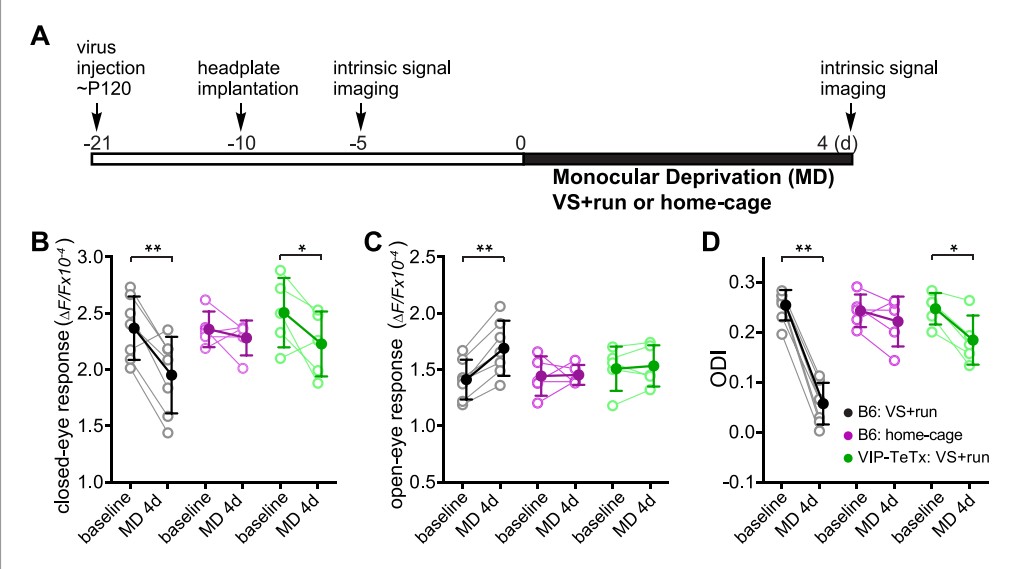

**Figure 2**. Synaptic transmission from VIP neurons facilitates the enhancement of ocular dominance plasticity in adult mice by running. (**A**) Experimental schedule. (**B** and **C**) Amplitudes of intrinsic signal responses evoked by the visual stimulation through the closed eye (**B**) and the open eye (**C**) before (baseline) and after 4-day monocular deprivation of the contralateral eye (MD 4d). (**D**) Ocular dominance index (ODI) computed from response amplitude to contralateral (closed) and ipsilateral (open) eyes shown in (**B** and **C**). Open circles represent measurements in individual animals. B6: C57BL/6J wild type mice (VS + run: n = 7; home-cage: n = 5); VIP-TeTx: VIP-Cre mice that received a cortical injection of AAV-DIO-TeTx and treated with VS + run during MD (n = 5). Solid circles represent the average of the corresponding open circles or open triangles (±S.D.). *p < 0.05, **p < 0.01 between baseline and after MD 4d; two-way ANOVA followed by multiple comparisons with Bonferroni correction.

and connected the cannula to a blue LED light source (20 Hz pulses, 2 s on, 1 s off, 4 hr each day) while the mice were housed in their home cages. No behavioral response to the optogenetic stimulus was evident during delivery. After 5-day MD in conjunction with optogenetic stimulation, we found that the visual response of the open-eye was dramatically potentiated (1.39 ± 0.19 vs 2.28 ± 0.81, p < 0.05; 64 ± 53% fold increase; *Figure 3D*, blue traces), significantly shifting ODI (0.22 ± 0.03 vs 0.05 ± 0.08, p < 0.05; *Figure 3E*, blue traces), with no significant change in the visual response of the closed eye (2.08 ± 0.12 vs 2.10 ± 0.55, p > 0.05; *Figure 3C*, blue traces). To rule out the possibility that the surgery procedure itself leads to enhanced plasticity, we injected VIP-Cre mice with AAV-DIO-TdTomato and subjected these control animals to the same procedure of optogenetic stimulation and 5-day MD (VIP-Tdtm group); no change was found in the open-eye response (1.77 ± 0.37 vs 1.62 ± 0.31, p > 0.05) and ODI (0.19 ± 0.06 vs 0.21 ± 0.05, p > 0.05; *Figure 3C–E*, red traces). Because AAV-DIO-ChETA may have labeled a small number of non-VIP neurons (*Pi et al., 2013*), we also examined whether stimulating non-specifically labeled, mostly pyramidal neurons would enhance adult plasticity. We injected AAV-ChETA into VIP-Cre mice and subjected the mice to the same procedure of optogenetic stimulation (pyr-ChETA group). We found no change in the visual responses to the two eyes (open eye: 1.89 ± 0.37 vs 1.78 ± 0.30, p > 0.05; closed eye: 2.64 ± 0.28 vs 2.36 ± 0.18, p > 0.05) or ODI (0.20 ± 0.02 vs 0.19 ± 0.02, p > 0.05) after 5-day MD (*Figure 3C–E*, magenta traces). These results therefore strongly support the hypothesis that activating VIP neurons is sufficient to enhance adult plasticity in the home cage without exercise on the trackball.

Activating VIP neurons is thought to enhance visual responses by inhibiting SST neurons and thereby disinhibiting pyramidal neurons (*Pfeffer et al., 2013*; *Fu et al., 2014*). We hypothesized that silencing SST neurons directly would be as effective as activating VIP neurons for enhancing adult plasticity. To test this hypothesis, we silenced SST neurons by injecting AAV-DIO-TeTx into adult SST-Cre mice (*Figure 4A*) and examined visual responses and ODI after 5-day MD (*Figure 4B*).

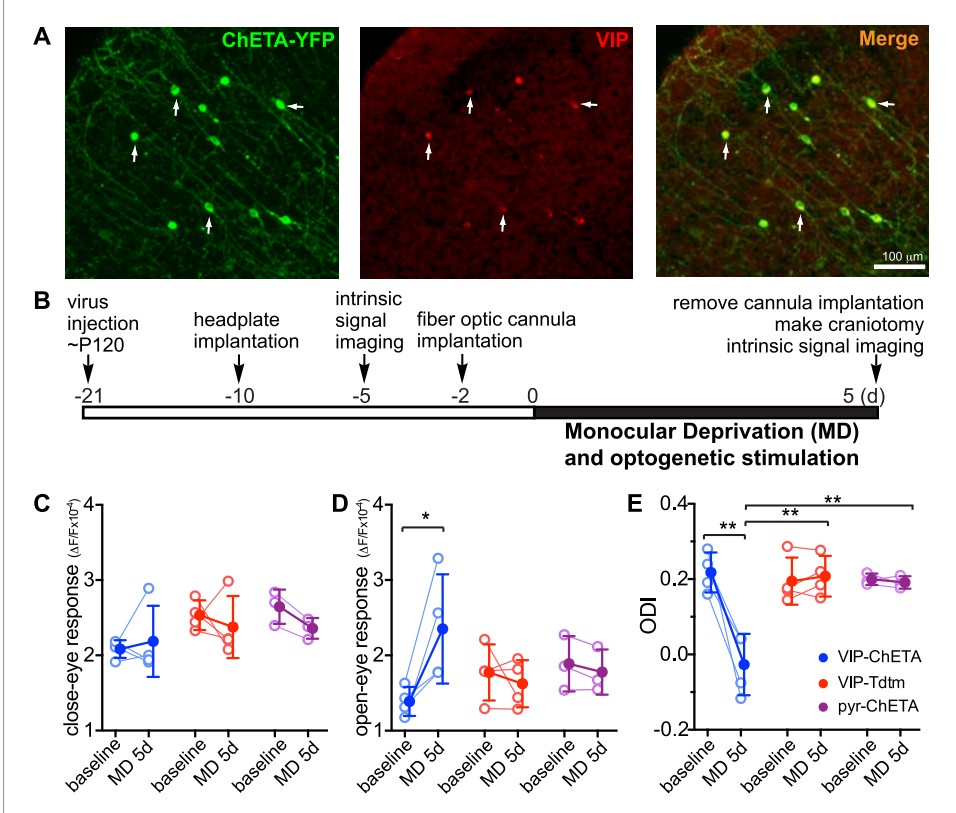

**Figure 3**. Activation of VIP neurons is sufficient to enhance visual cortical plasticity in adult mice. (**A**) Representative fluorescent images of binocular V1 area from a VIP-Cre mouse injected with AAV-DIO-ChETA-YFP. The slices were immunostained for YFP to indicate viral infected neurons and VIP for VIP-positive neurons. White arrows indicate the cells positive for both YFP and VIP staining. (**B**) Experimental schedule. (**C** and **D**) Changes in intrinsic signal responses evoked by the visual stimulation through the closed eye (**C**) and the open eye (**D**). VIP-Cre mice were injected with AAV-DIO-ChETA-YFP (VIP-ChETA, n = 4), AAV-DIO-TdTomato (VIP-Tdtm, n = 4), or AAV-ChETA (pyr-ChETA, n = 3). (**E**) Ocular dominance index (ODI) computed from response amplitude to contralateral (closed) and ipsilateral (open) eyes shown in (**C** and **D**). Open circles represent measurements in individual animals, and solid circles indicate mean of the open circles. (Data are plotted as mean ± S.D., *p < 0.05, **p < 0.01; paired t-test for comparing baseline and MD 5d of the VIP-ChETA group; other comparisons were analyzed with the two-way ANOVA followed by multiple comparisons with Bonferroni correction).

We found that the open-eye response in SST-Cre mice injected with AAV-DIO-TeTx (TeTx group) was significantly larger than that of either SST-Cre mice injected with AAV-DIO-TdTomato (Tdtm group) or C57BL/6J mice (B6 group) (TeTx: 1.92 ± 0.35; TdTomato: 1.42 ± 0.19; B6: 1.45 ± 0.27; *Figure 4D*), while the deprived-eye responses were indistinguishable between the three groups of mice (*Figure 4C*). Consequently, the ODI of TeTx group after 5-day MD was significantly different from those of the control Tdtm and B6 groups (TeTx: 0.06 ± 0.06, Tdtm: 0.22 ± 0.06, B6: 0.20 ± 0.07; *Figure 4E*). We also measured the changes in visual responses in a group of SST animals injected with AAV-DIO-TeTx before and after 5-day MD, yielding results that further confirmed that short-term silencing of SST neurons is sufficient to enhance adult plasticity by potentiating open-eye responses (*Figure 4—figure supplement 1*).

The VIP-SST disinhibitory circuit has recently been found to be a potent modulator of sensory responses and to regulate cortical states by integrating long-range inputs from other brain regions (*Pi et al., 2013*; *Fu et al., 2014*; *Zhang et al., 2014*). The present study reveals a new function of this disinhibitory circuit in regulating cortical plasticity.

Enhancing adult plasticity has long been a topic of great interest on account of its implications for human wellbeing. Inhibitory circuits have been implicated in regulating cortical plasticity. For example, both reducing GABA production and antagonizing GABA$_A$ receptors

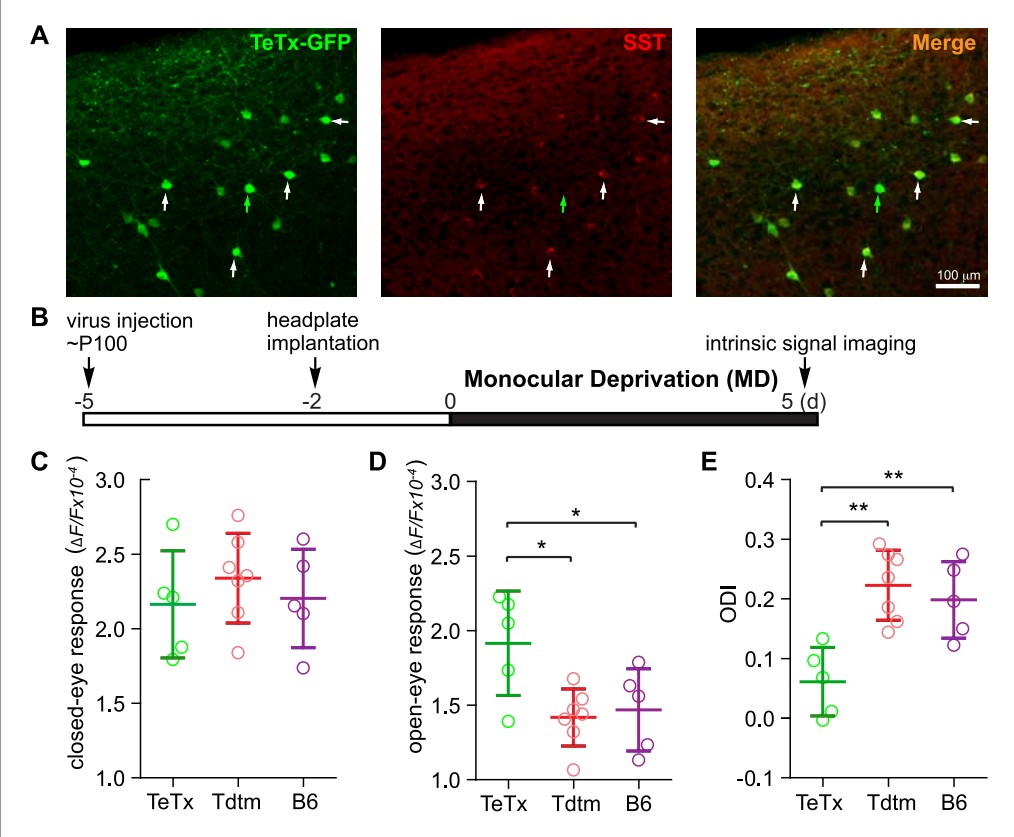

**Figure 4**. Short-term silencing of SST neurons is sufficient to enhance visual cortical plasticity in adult mice. (**A**) Representative fluorescent images of the binocular area in V1 from SST-Cre mouse injected with AAV-DIO-TeTx-GFP. The slices were immunostained for GFP to indicate viral infected neurons and SST for SST-positive neurons. White arrows indicate the cells positive for both GFP and SST staining, and green arrow indicates a cell positive only for GFP. (**B**) Experimental schedule. (**C** and **D**) Amplitudes of intrinsic signal responses evoked by visual stimulation through the closed eye (**C**) and the open eye (**D**), in SST-Cre mice treated with AAV-DIO-TeTx-GFP (TeTx, n = 5) or with AAV-DIO-TdTomato (Tdtm, n = 7) and in C57BL/6J mice (B6, n = 5). (**E**) Ocular dominance index (ODI) computed from response amplitude to contralateral (closed) and ipsilateral (open) eyes shown in (**C** and **D**). Open circles represent measurements in individual animals. (Data are plotted as mean ± S.D., *p < 0.05, **p < 0.01; One-way ANOVA followed by multiple comparisons with Bonferroni correction).

The following figure supplement is available for figure 4:

**Figure supplement 1**. Chronic intrinsic signal imaging confirms that the short-term silencing of SST neurons is sufficient to enhance visual cortical plasticity in adult mice by specifically potentiating open-eye response.

have been found to enhance adult plasticity in rat visual cortex, possibly by facilitating the potentiation of the input from the open eye (*Harauzov et al., 2010*). Chronic treatment using the antidepressant drug fluoxetine enhances adult plasticity, accompanied by reduced GABA levels in visual cortex (*Maya Vetencourt et al., 2008*). Our results are consistent with the idea that reduced inhibition is permissive for enhancing adult plasticity.

Removing extracellular perineuronal nets is also reported to enhance adult plasticity in rats (*Pizzorusso et al., 2002*; but see; *Vorobyov et al., 2013*). This and other findings suggest that simply destabilizing synaptic connections may enhance cortical plasticity, because reduced GABA transmission de-stabilizes GABAergic synaptic terminals (*Fu et al., 2012*). Transplanted GABAergic progenitor cells may create another form of instability of inhibitory circuits by making widespread new connections with host neurons and causing a second critical period of plasticity in adult mice (*Southwell et al., 2010*).

Extensive evidence has accumulated indicating that parvalbumin (PV)-positive fast-spiking cells play an important role in cortical plasticity (*Hensch, 2005*; *Kuhlman et al., 2013*). Our study reveals that disinhibiting pyramidal neurons by either activating VIP neurons or silencing SST neurons also enhances adult plasticity and allows potentiation of open-eye responses in V1, revealing an important role for at least these other major types of inhibitory neurons in cortical plasticity. Recent findings using transplantation of embryonic inhibitory neurons also support a role for SST neurons in cortical plasticity (*Tang et al., 2014*). It should be noted that there are multiple classes of SST neurons (*Markram et al., 2004*; *Hu et al., 2013*), only some of which are suppressed during locomotion (*Fu et al., 2014*; *Reimer et al., 2014*). In the present experiments, TeTx was presumably not selective in silencing the different subgroups of SST neurons.

Enhancing serotonin signaling by chronic fluoxetine treatment has also been shown to enhance adult plasticity (*Maya Vetencourt et al., 2008*). Exposure to an enriched environment was reported to enhance serotonin levels in rat visual cortex as well as to enhance adult plasticity (*Baroncelli et al., 2010*). Unlike PV and SST neurons, VIP neurons express the 5-HT$_3$ serotonin receptor, a channel that allows fast depolarization of neurons (*Lee et al., 2010*). VIP neurons also receive a direct nicotinic cholinergic input from the basal forebrain, which modulates cortical state and sensory responses (*Lee and Dan, 2012*; *Fu et al., 2014*). Therefore, the VIP-SST disinhibitory circuit is well poised as a target for manipulation of cortical plasticity by these two neuromodulators.

Adult ocular dominance plasticity in V1 consists of a potentiation of the open-eye response following MD; while critical period plasticity features an initial depression of the deprived-eye response followed by potentiation of the response to the open eye (*Sato and Stryker, 2008*). A number of manipulations in older animals mimic critical period plasticity in their effects on responses to the two eyes: transplantation of embryonic inhibitory neurons (*Southwell et al., 2010*) or acute suppression of the activity of PV neurons (*Kuhlman et al., 2013*) in mice, or chronic fluoxetine treatment in rats (*Maya Vetencourt et al., 2008*). Other manipulations, such as brief dark exposure (*He et al., 2006*; *Stodieck et al., 2014*), repeated MD (*Hofer et al., 2006*), visuomotor experience (*Tschetter et al., 2013*), or activation of Rho GTPases (*Cerri et al., 2011*), lead to a potentiation of open-eye responses that resembles normal adult plasticity following more prolonged MD. The present findings reveal that activation of the VIP-SST disinhibitory pathway is both necessary (*Figure 1, 2*) and sufficient (*Figures 3, 4*) for cortical plasticity that potentiates visual responses, and that aerobic exercise is neither necessary nor sufficient. Running, on the other hand, produces both potentiation of open eye responses and depression of responses to the closed eye. Running is associated not only with activation of cortical VIP neurons, but also with increases in multiple neuromodulators including serotonin and noradrenalin (*Meeusen and De Meirleir, 1995*), both of which have been implicated in modulating adult plasticity (*Gu, 2002*). It is, therefore, not surprising that enhancement of adult plasticity by locomotion is more complex than simply activating VIP-SST disinhibitory circuit, even though silencing VIP transmission abolished the effect of running on enhancing visual response (*Figure 1—figure supplement 1*) (*Fu et al., 2014*). More importantly, the present results also suggest that the potentiation of open-eye response and the depression of closed-eye response are separable from each other and are mediated by distinct mechanisms.

The difference between the plasticity produced by reducing the activity of PV- (*Kuhlman et al., 2013*) or SST- (*Figure 4*) neurons during MD suggests that different inhibitory circuits are engaged in gating distinct aspects of cortical plasticity. PV- and SST-cells innervate different compartments of pyramidal neurons and exert distinct physiological effects (*Markram et al., 2004*; *Hu et al., 2014*). For example, inhibitory synapses on dendritic spines (*van Versendaal et al., 2012*), which are preferentially innervated by SST neurons, are specifically lost during potentiation of the open-eye response in adult plasticity (*Chiu et al., 2013*). The different responses to reduced inhibition of distinct elements of the cortical circuit may be fundamental to understanding the differences between critical period and adult plasticity.

## Materials and methods

### Animals and monocular deprivation

VIP-Cre (stock No. 010908), SST-Cre (stock No. 013044), and C57B/L6 mice were from Jackson Lab. Experiments were performed on adult (age 3–6 months) mice of both sexes. The animals were

maintained in the animal facility at University of California, San Francisco (UCSF) and used in accordance with protocol AN098080-02A-G approved by the UCSF Institutional Animal Care and Use Committee. Animals were maintained on a 14 hr light/10 hr dark cycle. Experiments were performed during the light phase of the cycle.

Monocular deprivation (MD) was performed as described previously (*Gordon et al., 1996*) except that 2–3% isoflurane in oxygen was used for anesthesia. For long-term monocular deprivation, the lid of the right eye was sutured shut at P22–24. Mice were housed in the standard condition until P135-150, at which time a custom stainless steel plate for head fixation was attached to the skull with dental acrylic under isoflurane anesthesia. The exposed surface of the skull was covered with a thin coat of nitrocellulose (New-Skin, Medtech Products Inc., NY) to prevent desiccation, reactive cell growth, and destruction of the bone structure. Animals were given a subcutaneous injection of carprofen (5 mg/kg) as a post-operative analgesic. 5–7 days after head-plate implantation, the closed eyelid was re-opened and groups of mice for the intrinsic signal imaging study underwent imaging. The re-opened eyelid was left open afterward to allow binocular vision while animals were subjected to the visual stimulation and allowed locomotion as described previously (*Kaneko and Stryker, 2014*).

## Virus and viral injection

AAV-DIO-TeTx (AAV$_{DJ}$-DIO-TeTxLc-T2A-GFP) was a gift from Dr Wei Xu and Dr Thomas Sudhof at Stanford University (*Xu and Sudhof, 2013*). AAV-DIO-TdTomato (AAV$_{2.9}$-CAG-LSL-TdTomato) (Cat. No. AV-9-ALL856), AAV-DIO-ChETA (AAV$_{2.9}$-EF1a-DIO-ChETA-YFP) (Cat. No. AV-9-26968P), and AAV-ChETA (AAV$_{2.2}$-hSyn.ChETA) (Cat. No. AV-9-2-26967M) were purchased from University of Pennsylvania Vector Core, and injected as previously described (*Fu et al., 2014*). For viral injection, a small burr hole was drilled into the skull using a dental drill over the binocular zone of primary visual cortex in anesthetized mice. A glass micropipette (tip size ~10–30 μm) attached to a Picospritzer was lowered below the pia surface to the specified coordinates. 0.5–1 μl of the virus was injected with short pulses (50 ms) over 5 min. The glass pipette was left in place for an additional 3 min to allow viral diffusion. After removal of the injection pipette, the scalp was closed with vetbond (3M) and the animals were allowed to recover.

## Tetrode recording in awake mice

Our spherical treadmill was modified from the design described in *Niell and Stryker (2010)*. Briefly, a 200mm diam. closed-cell foam ball (Graham Sweet Studios, Cardiff, UK) was placed on a 250mm diam. foam bowl base with a single air inlet at the bottom. The base foam bowl was trimmed to allow the close placement of 2 USB-optical mice for sensing the rotation of the floating styrofoam ball, which transmitted the USB signals to our data analysis system. The animal's head was fixed via a surgically attached steel headplate screwed into a rigid crossbar above the floating ball. Recordings were performed as described previously (*Niell and Stryker, 2010*). On the day of recording, the animal was anesthetized with isoflurane in oxygen (3% induction, 1.5–2% maintenance). The skull was thinned above the area of viral infection so that it was nearly transparent. A small opening in the skull was made with a 27 gauge needle to allow insertion of a 16 channel probe (Neuronexus model a2X2-tet-2mm-150-121). The electrode was placed at an angle of ~45 deg relative to the cortical surface, to increase the distance between the insertion and recording sites. The electrodes were inserted to a depth of <400 μm below the cortical surface to record cells in layer 2/3. For each animal, the electrode was inserted only once. The animals recovered for >3 hr after craniotomy before recording data were collected.

## Visual stimulation, data acquisition, and analysis

Visual stimuli were presented as described previously (*Niell and Stryker, 2008*). Briefly, stimuli were generated in MatLab using the Psychophysics Toolbox extensions (*Brainard, 1997*; *Pelli, 1997*) and displayed with gamma correction on a monitor (Dell, 30 × 40 cm, 60 Hz refresh rate, 32 cd/m$^2$ mean luminance) placed 25 cm from the mouse, subtending ~60–75° of visual space. For drifting sinusoidal gratings, the spatial frequency was 0.05 cpd and the temporal frequency was 1 Hz. Stimulation was presented at nominal 100% contrast for 3 s with 0.5 s gray interval. For recovery from long-term MD, the visual stimulus was contrast modulated Gaussian noise, with a randomly generated spatiotemporal spectrum having low-pass spatial and temporal cutoffs applied at 0.05 cpd and 4 Hz, respectively, as previously described (*Kaneko and Stryker, 2014*). To provide contrast modulation, the noise pattern

was multiplied by a sinusoid with a 10-s period. Movies were generated at $60 \times 60$ pixels and then smoothly interpolated by the video card to $480 \times 480$ to appear $30 \times 30$ cm on the monitor and played at 30 frames per second. Each movie was 5 min long and repeated for 4 hr total presentation.

Data acquisition was performed as described by Niell and Stryker (2008). Signals were acquired using a System 3 workstation (Tucker–Davis Technologies, Alachua, FL). For single-unit activity, the extracellular signal was filtered from 0.7 to 7 kHz and sampled at 25 kHz. Spiking events were detected by voltage threshold crossing, and a 1 ms waveform sample on all four recording surfaces of the tetrode was acquired around the time of threshold crossing. Single-unit clustering and spike waveform analysis were performed as described previously (Niell and Stryker, 2008), using Klusta-Kwik (Harris et al., 2000, available at http://git.debian.org). Quality of separation was determined based on the Mahalanobis distance and L-ratio (Schmitzer-Torbert et al., 2005) and evidence of clear refractory period. Units were also checked to assure that their visual responses were similar at the beginning and end of recording to ensure that they had not drifted or suffered mechanical damage. As previously described (Niell and Stryker, 2008), units were classified as narrow or broad spiking based on properties of their average waveforms, at the electrode site with largest amplitude.

For drifting gratings, responses at each orientation were calculated by averaging the spike rate during the 3 s presentation and subtracting the spontaneous rate. The preferred orientation $\theta_{pref}$ was determined by averaging the response across all spatial frequencies, and calculating half the complex phase of the value $\sum F(\theta)e^{2i\theta}/\sum F(\theta)$. The orientation tuning curve was fitted as the sum of two Gaussians centered on $\theta_{pref}$ and $\theta_{pref} +\pi$, of different amplitudes but equal width, with a constant baseline.

## Optical imaging of intrinsic signals

After the headplate implantation, the first imaging of intrinsic signals was performed to measure baseline responses through each eye. The mouse was anesthetized with isoflurane (3% for induction and 0.7% during recording) supplemented with intramuscular injection of chlorprothixene chloride (2 µg/g body weight), and the closed eyelid was carefully opened by slitting horizontally at the center of the fused lid just before the imaging session. Repeated optical imaging of intrinsic signals was performed as described (Kaneko et al., 2008). We monitored the concentration of isoflurane using an Ohmeda 5250 RGM (Datex-Ohmeda, Madison, WI) throughout each imaging session. Images were recorded transcranially through the window of the implanted headplate. Intrinsic signal images were obtained with a Dalsa 1M30 CCD camera (Dalsa, Waterloo, Canada) with a $135 \times 50$ mm tandem lens (Nikon Inc., Melville, NY) under illumination with a red LED (measured peak emission 612 nm, part LXHL-NX05, The LED Light, Carson City, NV, discontinued; similar to SP-12-E4 plus FLP-N4_RE-HRF from Quadica Developments Inc, Brantford, ONT, Canada). Frames were acquired at a rate of 30 fps, temporally binned by four frames, and stored as $512 \times 512$ pixel images after binning the $1024 \times 1024$ camera pixels by $2 \times 2$ pixels spatially. The visual stimulus used to record intrinsic signals from binocular V1 was the contrast-modulated noise movie described above, presented between $-5°$ and $15°$ (azimuth) on a $40 \times 30$ cm monitor placed 25 cm in front of the mouse ($0°$ = center of the monitor aligned to center of the mouse). The phase and amplitude of cortical responses at the stimulus frequency were extracted by Fourier analysis as described (Kalatsky and Stryker, 2003). Response amplitude was an average of at least four measurements. Ocular dominance index was computed as $(R - L)/(R + L)$, where R and L are the peak response amplitudes through the right eye and the left eye, respectively, as described (Kaneko et al., 2008). All mice were kept under standard housing conditions with free access to food and water between recordings and daily running on the treadmill in indicated experiments.

## Fiber optic cannula implantation and optogenetic stimulation

3 weeks after viral injection, a cannula with optic fiber in the center (0.2 mm in diameter for the optical fiber) (Thorlabs, CFMC12U-20; the optic fiber was cut to be 1 mm in length) was implanted on top of the center of the binocular zone and secured to the skull with dental cement. A 470 nm LED (Thorlabs, M470F1) with 400 µm diameter core fiber output was used to generate blue light. The fiber optic inserted into the cannula was illuminated by the LED through a flexible optic fiber with rotatory joint that allowed the animal to move freely in its home-cage, at a final intensity emitted from the fiber of 15 mW/mm². Light pulse stimuli (2 s of 20 Hz (25 ms on, 25 ms off) and 1 s off) were generated by

connecting the LED controller with a Master-8 pulse stimulator. The mice were given daily optogenetic stimulation for 4 hr per day for 5 consecutive days.

## Immunohistochemistry

Animals were transcardially perfused with saline and 4% formaldehyde. The brains were removed, cryoprotected in 30% sucrose, and cut into 30 μm coronal sections on a frozen sliding microtome (Physitemp Instruments). Floating sections were blocked for 1 hr at room temperature in Tris buffered saline (TBS) containing 10% normal goat serum and 1% Tween-100, then incubated overnight at 4°C using in the same solution with the following antibodies: chicken anti-GFP, 1:1000 (Aves); rabbit anti-SST, 1:300 (Swant); and rabbit anti-VIP, 1:1000 (Immunostar). The sections were then washed in TBS with 1% Tween-100 three times for 15 min, incubated for 1 hr at room temperature in blocking solution with 1:1000 each of Alexa Fluor 488 goat anti-chicken and Alexa Fluor 568 goat anti-rabbit (Life Technologies). The sections were washed in phosphate buffered saline three times for 10 min, mounted on glass slides, dried, and covered with coverslips. Images of the visual cortex were captured using a Zeiss Axiovert-200 micrscope and AxioCam Mrm (Zeiss). Contrast was adjusted in ImageJ.

## Acknowledgements

We thank Drs Wei Xu and Thomas Sudhof for providing AAV-DIO-TeTx virus. This work was supported by NIH grants R01EY02874 (MPS), 5T32EY007120 (YF) and T32MH089920 (MK) and a grant to MPS from the Simons Collaboration on the Global Brain.

## Additional information

### Funding

| Funder | Grant reference number | Author |
|---|---|---|
| National Institutes of Health | R01EY02874 | Michael P Stryker |
| National Institute of Mental Health | T32MH089920 | Megumi Kaneko |
| National Institutes of Health | 5T32EY007120 | Yu Fu |

The funders had no role in study design, data collection and interpretation, or the decision to submit the work for publication.

### Author contributions

YF, MK, Conception and design, Acquisition of data, Analysis and interpretation of data, Drafting or revising the article; YT, Acquisition of data, Drafting or revising the article; AA-B, Drafting or revising the article; MPS, Conception and design, Analysis and interpretation of data, Drafting or revising the article

### Ethics

Animal experimentation: Mice were maintained in the Laboratory Animal Research Center at University of California, San Francisco (UCSF) and used in accordance with protocol AN098080-02A-G approved by the UCSF Institutional Animal Care and Use Committee.

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
