## [Decision Letter]

Thank you for sending your work entitled “A cortical disinhibitory circuit for enhancing adult plasticity” for consideration at *eLife*. Your article has been favorably evaluated by Eve Marder (Senior editor) and two reviewers, one of whom, Sacha Nelson, is a member of our Board of Reviewing Editors.

The Reviewing editor and the other reviewer discussed their comments before we reached this decision, and the Reviewing editor has assembled the following comments to help you prepare a revised submission.

The authors and colleagues have previously shown that exercise (running) enhances visual responses through a disinhibitory circuit involving VIP and SST interneurons and that exercise enhances the recovery from amblyopia produced by monocular deprivation (prior *eLife* paper). The present study tests the idea that the activation of the disinhibitory circuit is necessary and sufficient for the enhanced plasticity during recovery from monocular deprivation, and finds convincingly that the two sets of phenomena are causally related. Other aspects of exercise (e.g. stimulation of neurogenesis, activation of other neuromodulatory pathways etc.) have previously been implicated in the effect of exercise on plasticity, so the present results were not a foregone conclusion.

The authors first show that VIP activity is necessary for the increased ocular dominance plasticity that is observed in adult animals that are running while viewing stimuli. They also show that activating these VIP neurons intensifies ocular dominance plasticity in normally-reared mice that have been monocularly deprived as adults. The authors then show that VIP activity is sufficient to enhance ocular dominance plasticity in adult mice, using optogenetics. Finally, the authors hypothesize that these VIP neurons are acting to inhibit SST neurons; therefore, inactivating SST neurons should also enhance plasticity. The authors perform this experiment in Figure 3, and show that this enhances plasticity in adult mice.

This represents a significant advance, but one which is tightly linked to the prior paper and so is quite appropriate for a Research Advance. The experiments appear to have been carefully carried out and convincingly controlled and both reviewers had no major concerns.

Both reviewers also felt that the function of different classes of interneurons in visual cortical plasticity are of great general interest, and expect that this paper will be widely read.

Minor comments:

1) Were there any acute behavioral effects of the silencing and activating reagents used? The control for amount of running is convincing but it would also be helpful to note the presence or absence of any obvious changes in visual behavior, anxiety, etc. and if present, why these are unlikely to have contributed to the observed effects.

2) The removal of perineuronal nets is relatively controversial (see [42], J Neurosci, Jan 2;33(1):234–43.). “Some reports suggest that removing extracellular perineuronal…”

---

## [Author Response]

In the revised manuscript, we have addressed all the reviewers’ comments and performed more experiments to further strengthen our conclusions. We showed in the original manuscript that running enhanced the shift of ocular dominance index after 4-day monocular deprivation in adult mice, by both depressing the closed-eye response and potentiating the open-eye response. Our new results show that the synaptic transmission of VIP neurons is specifically involved in potentiating the open-eye response (new Figure 2; original Figure 2 and Figure 3 are now Figure 3 and Figure 4, respectively), which is the feature characteristic of normal adult ocular dominance plasticity after more prolonged periods of deprivation. Furthermore, in another group of mice, we did longitudinal intrinsic signal imaging, studying individual animals before and after deprivation, to confirm that the open-eye response is potentiated after 5-day monocular deprivation when SST neurons are silenced (now Figure 4—figure supplement 1). We also revised our manuscript accordingly to include these new findings.

Minor comments:

*1) Were there any acute behavioral effects of the silencing and activating reagents used? The control for amount of running is convincing but it would also be helpful to note the presence or absence of any obvious changes in visual behavior, anxiety, etc. and if present, why these are unlikely to have contributed to the observed effects*.

We did not observe any acute behavioral effects of silencing and activating VIP or SST neurons. In the revised manuscript, we added the following sentences: “In these animals, as in all of the TeTx-treated animals described below, we noted no obvious changes in temperament or visual behavior; experimental and control animals appeared indistinguishable” (paragraph five), and “No behavioral response to the optogenetic stimulus was evident during delivery” (paragraph seven).

*2) The removal of perineuronal nets is relatively controversial (see*
[42]*, J Neurosci, Jan 2;33(1):234–43.). “Some reports suggest that removing extracellular perineuronal…*”

We revised the related text to be “Removing extracellular perineuronal nets is also reported to enhance adult plasticity in rats ([32]; but see [42]).”